# Identification and Characterisation of Stripe Rust Resistance Genes *Yr66* and *Yr67* in Wheat Cultivar VL Gehun 892

Harbans Bariana [1,*] , Lakshmi Kant [1,2] , Naeela Qureshi [1,3] , Kerrie Forrest [4], Hanif Miah [1] and Urmil Bansal [1,*]

1   Plant Breeding Institute, School of Life Sciences, Faculty of Science, The University of Sydney, 107 Cobbitty Road, Cobbitty, NSW 2570, Australia; lkant_vpkas@yahoo.com (L.K.); n.qureshi@cgiar.org (N.Q.); hanif.miah@sydney.edu.au (H.M.)
2   Indian Council of Agricultural Research (ICAR)-Vivekananda Parvatiya Krishi Anusandhan Sansthan (VPKAS), Almora 263601, Uttarakhand, India
3   International Maize and Wheat Improvement Center (CIMMYT), El Batán, Texcoco 56237, Mexico
4   Agriculture Victoria Research, Department of Economic Development, Jobs, Transport and Resources, AgriBio, Bundoora, VIC 3083, Australia; kerrie.forrest@agriculture.vic.gov.au
*   Correspondence: harbans.bariana@sydney.edu.au (H.B.); urmil.bansal@sydney.edu.au (U.B.)

**Abstract:** Wheat cultivar VL Gehun 892 has shown a high level of resistance against Australian *Puccinia striiformis* f. sp. *tritici* (Pst) pathotypes. In this study, it was crossed with Westonia, a susceptible wheat cultivar, and digenic segregation was observed in the derived population against Pst pathotype 134 E16A+Yr17+Yr27+. Single-gene recombinant inbred line (RIL) populations were developed from F3 families (VL Gehun 892/Westonia#1 and VLGehun 892/Westonia#4) that showed monogenic segregations with two distinct phenotypes. Single-gene segregation against Pst pathotype 134 E16A+Yr17+Yr27+ was confirmed in both F6 RIL populations. Bulked segregant analysis using a 90K Infinium SNP array placed *YrVL1* in the short arm of chromosome 3D and *YrVL2* in the long arm of chromosome 7B. Kompetitive allele specific polymerase chain reaction (KASP) assays were developed for the SNPs linked with *YrVL1* and *YrVL2* and were mapped onto the respective populations. *KASP_48179* (0.6 cM proximal) and *KASP_18087* (2.1 cM distal) flanked *YrVL1*, whereas *YrVL2* was mapped between *KASP_37096* (1.2 cM proximal) and *KASP_2239* (3.6 cM distal). Based on their pathotypic specificities, map locations, and stages of expression, *YrVL1* and *YrVL2* were demonstrated to be unique loci and named *Yr66* and *Yr67*, respectively. Markers linked with these genes showed more than 85% polymorphism when tested on a set of 89 Australian cultivars and hence could be used for the marker-assisted selection of these genes in wheat breeding programs, following checks of parental polymorphisms.

**Keywords:** KASP markers; marker-assisted selection; stripe rust; wheat; *Yr* genes

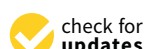


## 1. Introduction

The causal agents of rust diseases in the *Puccinia* species of wheat were ranked among the top 10 fungal pathogens in a survey of plant pathologists associated with the journal *Molecular Plant Pathology* [1]. Stripe rust caused by *Puccinia striiformis* f. sp. *tritici* (Pst) was estimated to cause A\$127 million in losses in Australia [2]. The deployment of resistance is the preferred option for stripe rust control due to environmental and economic considerations [3–5]. Genetic control can be achieved by deploying race-specific all-stage resistance (ASR) and non-race-specific adult plant resistance (APR) genes [3,6–8]. The deployment of a single ASR gene in commercial cultivars often leads to the breakdown of resistance due to the acquisition of virulence in the pathogen population [9,10].

Although new pathotypes with virulence for stripe rust resistance genes that were previously effective against the predominant pathotypes have appeared recently (pathotypes 239 E139A-Yr17+Yr33+ and 198 E16A-Yr17+YrJ+YrT+), these pathotypes carried avirulence for some previously ineffective genes. For example, the new pathotype 239

E139A-Yr17+Yr33+ possessed virulence for *Yr33*, *Yr57*, *Yr63*, *Yr72*, and *Yr75*, whereas it was avirulent for the dominant complementary gene *YrA* (H.S. Bariana unpublished results). Of these, *Yr57* and *Yr63* have not even been deployed in any commercial cultivar of wheat yet. This pathotype is also clearly avirulent for the stripe rust resistance gene *Yr27*, which has led to epidemics in several parts of the world. These features of pathotypic evolution suggest that we should develop markers closely linked with the currently deployed stripe rust resistance genes in international cultivars that are effective against either one or several pathotypes. Hence, releasing wheat cultivars with combinations of ASR genes that show differential pathotypic specificities with APR genes is the better strategy to avoid future epidemics. Combinations of genes with compensating pathotypic specificities are conditioning stripe rust resistance in different geographic regions around the globe. In many cases, the inheritance of stripe rust resistance remains poorly understood. Advances in wheat genomics, such as the development of various genotyping arrays (9K, 90K, DArTseq, etc.) and the availability of reference whole-genome sequences of wheat, have expedited discovery of genetically diverse sources of resistance. These developments have also led to the identification of close marker-trait associations. In many cases, allele specific assays based on single-nucleotide polymorphisms have been developed to facilitate the marker-assisted pyramiding of genes.

An Indian wheat genotype, VL Gehun 892, expresses stripe rust resistance under both field and greenhouse conditions. It produced infection type (IT) ;CN under greenhouse conditions when tested against the Pst pathotype 134 E16A+Yr17+Yr27+. We hypothesised that VL892 either carries a new stripe rust resistance gene or a combination of genes with compensating pathotypic specificities. The screening of VL892/Westonia-derived $F_3$ families against the Pst pathotype 134 E16A+Yr17+Yr27+ indicated the digenic inheritance of seedling stripe rust resistance (H.S. Bariana unpublished results). This manuscript covers the identification of the chromosomal locations of loci conferring resistance in VL Gehun 892 and the development of markers closely linked with each locus for marker-assisted pyramiding in wheat breeding programs.

## 2. Materials and Methods

One hundred seeds of VL Gehun 892/Westonia-derived and monogenically segregating $F_3$ families, VL Gehun 892/Westonia#1 (phenotype 1) and VL Gehun 892/Westonia#4 (phenotype 2), were sown and a single head from each plant was harvested to generate single-gene segregating recombinant inbred line (RIL) populations. The seed of the $F_5$ single-plant progeny was harvested to develop $F_6$ RILs. These populations were named VL/Wes#1 (80 RILs) and VL/Wes#4 (77 RILs). In addition, a set of 89 Australian wheat cultivars was used to test the polymorphism of resistance-linked markers.

### 2.1. Greenhouse Testing

Eight to ten seeds of each RIL from the VL/Wes#1 and VL/Wes#4 populations were sown in small pots in the greenhouse as four lines per pot, inoculated with Pst pathotype 134 E16A+Yr17+Yr27+, and scored using a 0–4 infection type (IT) scale [11]. This scale covers ITs 0, 1, 2, 3, and 4, where ITs 3 or lower are considered resistant and more than 3 are considered susceptible. Symbols '−' and '+' explained slight deviations from a usual IT, whereas more than usual chlorosis and necrosis was denoted by 'C' and 'N', respectively. The parents VL892 and Westonia were included as controls. These pots were moved to a rust-free and temperature-controlled greenhouse room. This material was inoculated with Pst pathotype 134 E16A+Yr17+Yr27+ in the greenhouse at the two-leaf stage to confirm single-gene segregation in the VL/Wes#1 and VL/Wes#4 populations, according to Bariana and McIntosh (11), and the RILs were classified as homozygous-resistant (HR) and homozygous-susceptible (HS).

Two HR lines each from VL/Wes#1 and VL/Wes#4, along with the susceptible control Morocco, were tested with four Pst pathotypes, 110 E143A+, 134 E16A+Yr17+Yr27+, 239

E237A-Yr17+Yr33+, and 198 E16A-J+T+Yy17+ following the procedure described in Bariana and McIntosh [11].

### 2.2. DNA Extraction

DNA was extracted from VL/Wes#1 and VL/Wes#4 RIL populations using the modified CTAB method described in Bansal and co-workers [12]. DNA was quantified using a Nanodrop ND-1000 spectrophotometer (Nanodrop Technologies, Wilmington, DE, USA).

### 2.3. Bulked Segregant Analysis (BSA)

Equal amounts of DNA of 20 HR and 20 HS lines were used to prepare resistant and susceptible bulks, respectively, from both populations. An artificial $F_1$ was created by pooling equal amounts of DNA from 40 random lines. The DNA bulks and artificial $F_1$ were subjected to BSA using the 90K iSelect Infinium SNP array. The single-nucleotide polymorphisms (SNPs) showing association with resistance in bulks from both populations through comparing normalized theta values were converted to kompetitive allele-specific polymerase chain reaction (KASP) assays and tested on four resistant and susceptible RILs each from both populations following the methodology described in Nsabiyera et al. [13].

### 2.4. Genetic Analysis and Linkage Mapping

Chi-squared analyses were conducted to test the goodness of fit of the observed segregation with expected genetic ratios among RIL populations. MapManager QTXb20 [14] and Kosambi mapping function [15] were used for linkage analysis. The MapChart software [16] was used to order markers in the final linkage map.

## 3. Results

### 3.1. Inheritance of Stripe Rust Resistance

Both VL/Wes#1 and VL/Wes#4 RIL populations were tested at the two-leaf stage with Pst pathotype 134E16A+Yr17+Yr27+. Monogenic segregation for stripe rust resistance in both populations was observed (Table 1). The stripe rust responses of HR lines from VL/Wes#1 ranged from IT 1C to 2C, whereas the HR lines from VL/Wes#4 exhibited IT 23C. The resistance loci conditioning stripe rust resistance in VL/Wes#1 and VL/Wes#4 were temporarily named *YrVL1* and *YrVL2*, respectively.

**Table 1.** Frequency distribution of VL/Wes#1 and RIL#4 RIL populations against *P. striiformis* f. sp. *tritici* pathotype 134 E16A+Yr17+Yr27+.

| Generation | Population | Rust Response * | No of Lines | | $\chi^2$ (1:1) |
| --- | --- | --- | --- | --- | --- |
| | | | Observed | Expected | |
| RIL | VL/Wes#1 | HR | 33 | 40.0 | 1.225 |
| | | HS | 47 | 40.0 | 1.225 |
| | | Total | 80 | 80.0 | 2.450 |
| | VL/Wes#4 | HR | 38 | 38.5 | 0.006 |
| | | HS | 39 | 38.5 | 0.006 |
| | | Total | 77 | 77.0 | 0.013 |

Table value of $\chi^2$ at *p* = 0.05 and 1 *df* = 3.84. * HR: homozygous-resistant; HS: homozygous-susceptible.

### 3.2. Molecular Mapping

#### 3.2.1. YrVL1

Thirty-four SNPs located in the short arm of chromosome 3D showed a strong association with *YrVL1* in the 90K SNP array-based BSA. These SNPs were converted into KASP assays. Out of 34 KASP markers, 11 (*KASP_44133*, *KASP_10496*, *KASP_23205*, *KASP_56281*, *KASP_29499*, *KASP_48159*, *KASP_18087*, *KASP_48179*, *KASP_6764*, *KASP_60455*, and *KASP_4059*) clearly differentiated parental genotypes and four homozygous resistant

and four homozygous susceptible RILs. These markers were tested on the VL/Wes#1 RIL population (Table 2). The final genetic linkage map for *YrVL1* consisted of 11 KASP markers covering a total genetic distance of 12.5 cM. *KASP_48179* and *KASP_18087* mapped 0.6 cM (proximal) and 2.1 cM (distal) to *YrVL1* (Figure 1).

**Table 2.** List of kompetitive allele specific polymerase chain reaction (KASP) markers used for the molecular mapping of *YrVL1* and *YrVL2*.

| Marker Name | Physical Position (bp) | Primer Sequence | | |
|---|---|---|---|---|
| | | Allele Specific Primer A1 [a] | Allele Specific Primer A2 [b] | Common Primer |
| *YrVL1* | | | | |
| *KASP_44133* | 2,242,884 | agaagaaagggtgccgccatga | gaagaaagggtgccgccatgc | ggcagccgatcgactacatgttatt |
| *KASP_10496* | 2,400,568 | agtggaggtcgaagcatgcagt | gtggaggtcgaagcatgcagc | gctgcccaggctggctcgta |
| *KASP_23205* | 2,593,371 | gacagcattgtCgataaacact | gacagcattgtCgataaacacg | ggtgccaccacaagttctta |
| *KASP_56281* | 2,626,318 | catcctcaaacttggtattatcagaagta | cctcaaacttggtattatcagaagtg | tcataaaggcactctacaacttgcttgat |
| *KASP_29499* | 2,981,592 | ccggaagaccgctgaccca | cggaagaccgctgacccg | ggttaatgttgccacttccatagtcaaa |
| *KASP_48159* | 3,237,190 | ccctgttcaagccagaggt | ccctgttcaagccagaggc | acctgcttgaaagagcggc |
| *KASP_18087* | 3,549,840 | aaagaaaggcctcagtttgacgatatt | aagaaaggcctcagtttgacgatatc | ggtcattgtcaagcgcttccttgta |
| *KASP_48179* | Not known | agtacatatccccagctgctt | agtacatatccccagctgctc | agctggacatggtgctgttt |
| *KASP_6764* | Not known | gttcataataaagaggaggctggt | gttcataataaagaggaggctggc | ttcagcgggaactcgaacat |
| *KASP_60455* | 3,301,402 | gttctctttgaccgccttaca | gttctctttgaccgccttacg | gCtCatctcagcgagatctca |
| *KASP_4059* | 2,976,737 | ggagataccgggtaaagaaatca | ggagataccgggtaaagaaatcg | aattaccaatattttgccatgtgc |
| *YrVL2* | | | | |
| *KASP_13220* | 712,082,664 | gaattttactgctgttattgtggtgttt | gaattttactgctgttattgtggtgttg | gccttcagcattacagcaaatgaaatgaa |
| *KASP_62475* | 716,963,586 | gttgaatgtctactttcatccgcca | gaatgtctactttcatccgccg | caggataatactgtgcaaaccagcctt |
| *KASP_37096* | 716,966,240 | gagcgccaacgcttggttcca | gcgccaacgcttggttccc | cttgacaagacacagtagggtatacatat |
| *KASP_2239* | 721,207,696 | ctctagggtggcgaggca | ctctagggtggcgaggcg | catacagctaaccgggcgag |
| *KASP_71995* | 721,207,752 | catggtggcgctctgatccgt | ggtggcgctctgatccgc | catacagtctgcttattaacatacagctaa |
| *KASP_61786* | 750,083,910 | cgcagttttgacacctcgt | cgcagttttgacacctcgc | caattcggagttgtagaattcttca |

[a] A1 primer labelled with FAM: GAAGGTGACCAAGTTCATGCT; [b] A2 primer labelled with HEX: GAAGGTCGGAGTCAACGGATT.

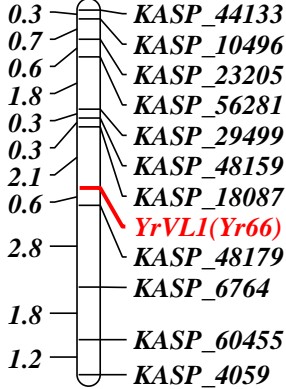

**Figure 1.** Genetic linkage map of chromosome 3DS showing the location of *Yr66*.

### 3.2.2. *YrVL2*

Of the polymorphic SNP in BSA, 20 markers were identified from the long arm of chromosome 7B that showed strong associations with *YrVL2*. These SNPs were converted into KASP assays and tested on parental genotypes for polymorphism. Six KASP markers, *KASP_37096*, *KASP_62470*, *KASP_13220*, *KASP_2239*, *KASP_71995*, and *KASP_61786*, showed different parental alleles and differentiated four homozygous resistant and four homozygous susceptible RILs and were tested on the VL/Wes#4 RIL population (Table 2). A genetic linkage map was constructed using six KASP markers spanning a genetic distance of 15.6 cM. *YrVL2* was flanked by *KASP_37096* and *KASP_2239* at 1.2 cM (proximal) and 3.6 cM (distal) distances (Figure 2).

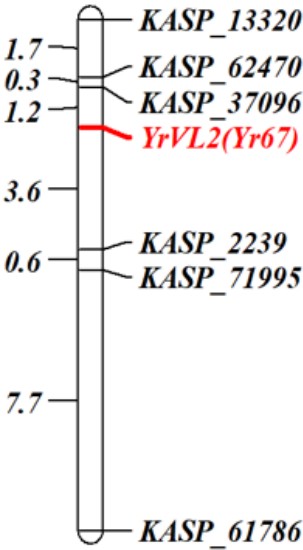

**Figure 2.** Linkage map of the long arm of chromosome 7B of VL/Wes#4 recombinant inbred line population showing location of *Yr67*.

### 3.2.3. Polymorphism of *YrVL1*- and *YrVL2*-Linked Markers

Markers that flanked *YrVL1* and *YrVL2* were tested on a set of 89 Australian wheat cultivars. *YrVL1*-linked markers *KASP_48179* and *KASP_18087* were polymorphic in 88 and 86% of tested cultivars, respectively (Table 3). Marker *KASP_37096* linked with *YrVL2* produced much less polymorphism (50%, data not given), while *KASP_2239* was polymorphic in 86% of the wheat cultivars. These markers can be used in the marker-assisted selection of these genes in wheat breeding following the determination of parental polymorphism.

**Table 3.** Validation of *Yr66*- and *Yr67*-linked kompetitive allele specific polymerase chain reaction (KASP) markers on a set of 89 Australian wheat cultivars.

| Cultivar/Stock. | KASP_2239 | KASP_18087 | KASP_48179 |
|---|---|---|---|
| *YrVL1* (*Yr66*) | - | *C:C* | *T:T* |
| *Yrvl1* (*yr66*) | - | *T:T* | *C:C* |
| *YrVL2* (*Yr67*) | *G:G* | - | - |
| *Yrvl2* (*yr67*) | *A:A* | | |
| | | - | - |
| Correll, Espada, Cobra, AGT Katana, Baxter, Calingiri, Carnamah, Catalina, Chara, Chief CL Plus, Coolah, Corack, Crusader, Dart, Derrimut, DS Faraday, EGA Bonnie Rock, EGA Burke, EGA Gregory, EGA Wedgetail, EGA Wylie, Emu Rock, Fortune, Gauntlet, Gazelle, GBA Sapphire, Giles, Gladius, Grenade CL Plus, Impala, Impose CL Plus, Janz, Justica CL Plus, Lancer, Lang, Livingston, LRPB Arrow, LRPB Flanker, LRPB Kittyhawk, LRPB Reliant, Magenta, Mansfield, Merinda, Naparoo, Preston, SF Scenario, Shield, Strzelecki, Sunco, Sunguard, Sunmax, Sunvale, Trojan, Waagan, Wallup, Westonia, Wyalkatchem, Wylah, Yandanooka | *A:A* | *T:T* | *C:C* |
| Axe, EGA Bounty, Lincoln, Sentinel, SF Adagio, Spitfire, Suntop, Sunzell, Ventura, Beaufort | *G:G* | *T:T* | *C:C* |
| Kunjin, Merlin | *G:G* | *C:C* | *C:C* |
| Elmore CL PLus, King Rock, Kord CL Plus, Mace, Mackellar, Ninja, Young | *A:A* | *C:C* | *C:C* |
| Orion, Phantom, SQP Revenue, | *A:A* | *C:C* | *T:T* |
| Bolac, Envoy, Forrest, Harper, Scout, Wedin, Yitpi | *A:A* | *T:T* | *T:T* |

### 3.3. Effectiveness of YrVL1 and YrVL2 against Different Pst Pathotypes

Homozygous-resistant lines from both populations were tested against four Pst pathotypes. *YrVL2* displayed resistance against all four pathotypes and the ITs varied from ;C to 23C (Figure 3). *YrV1* was effective against two (134 E16A+Yr17+Yr27+ and 198 E16A-J+T+Yr17+) of the four pathotypes tested (Figure 3).

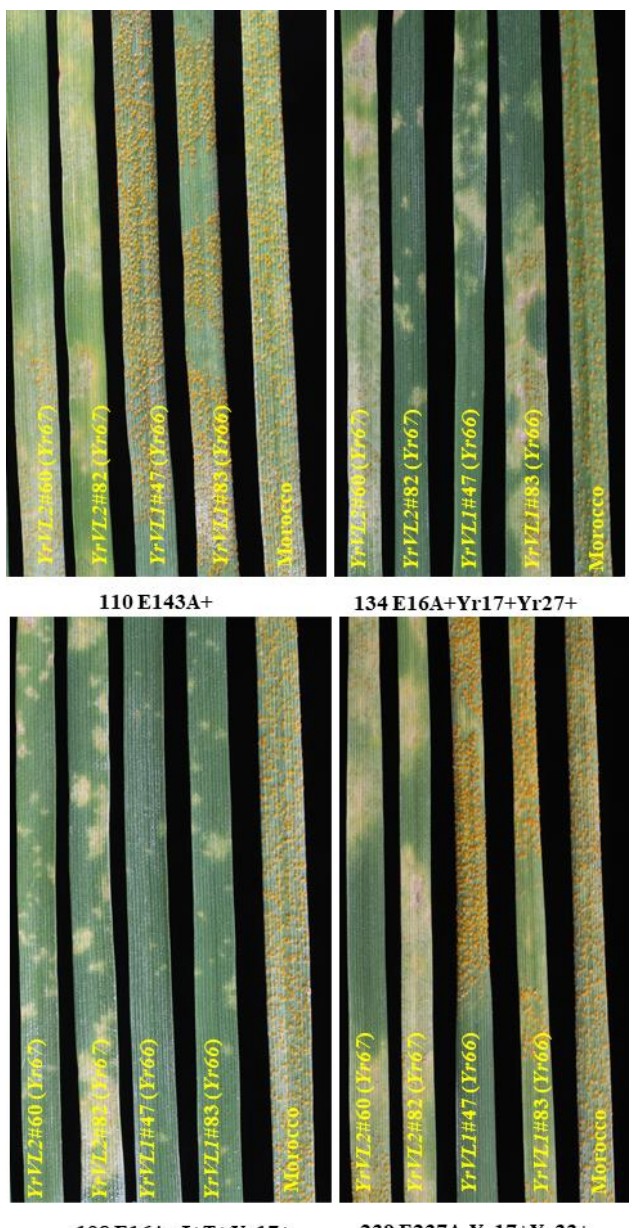

**Figure 3.** Infection types produced by two homozygous resistant RILs each from VL892/Westonia#4 (*Yr67*) and VL892/Westonia#1 (*Yr66*) populations and the susceptible control Morocco against four *P. striiformis* f. sp. *tritici* pathotypes.

## 4. Discussion

Continuous evolution in the stripe rust pathogen has restricted the availability of effective sources of stripe rust resistance for the development of cultivars with durable resistance [17]. This emphasises the need for the identification and characterisation of new sources of resistance effective against the stripe rust pathogen for successful utilisation in breeding programs. This study dissected resistance carried by cultivar VL Gehun 892 and the underlying loci, which were temporarily named *YrVL1* and *YrVL2*.

Tests on single-gene stocks showed that *YrVL1* was effective against two post-2002 Australian Pst pathotypes, whereas *YrVL2* showed resistance against all four pathotypes. *YrVL1* was flanked by *KASP_18087* (3,549,840 bp of the Chinese Spring reference genome) and *KASP_48179* (position not known). These SNP markers spanned an interval of 0–4.5 cM of the total 95 cM map length of chromosome 3DS [18]. An adult plant resistance gene *Yr49* has previously been reported in chromosome 3DS; flanked by markers *gpw7321* (12.5 cM) and *gwm161* (25 cM; 7,094,923 bp), in a total map length of 154 cM [19]. The comparison of map locations indicate that *Yr49* is proximal to *YrVL1*. Based on the different chromosomal locations and the fact that *YrVL1* is a seedling resistance gene, *YrVL1*, was permanently designated *Yr66*.

Molecular mapping using the 90K Infinium SNP array positioned *YrVL2* in the long arm of chromosome 7B (716,966,290 to 721,082,714 bp), and an old Indian cultivar C591 in which *YrC591* was mapped appears in the pedigree of VL Gehun 892. *YrC591* was flanked by SSR marker *cfa2040* and AFLP marker *SCP35M48* and was mapped in chromosome 7BL [20]. SSR markers *cfa2040* and *barc182* were both mapped at the 164 cM position in Somers et al. [19]. *YrVL2*-linked SNPs were genotyped on cultivars C591 and C306 and amplified the *YrVL2*-linked alleles. Based on pedigree information, the map positions of SSR markers, and the amplification of *YrVL2*-linked products in C591 and C306, we concluded that *YrVL2* and *YrC591* represent the same locus. The other stripe rust resistance genes that are mapped in chromosome 7BL include *Yr6* [21,22], *Yr39* [23], *Yr52* [24], and *Yr59* [25]. *Yr6* is linked with marker *gwm577* [19; position: 157 cM] and is not effective against the Pst pathotype used in this study. On the other hand, *Yr39* is a high-temperature adult plant (HTAP) resistance gene that mapped 7 cM proximal to the SSR marker *gwm131* [98 cM in 20]. *Yr52* was flanked by SSR markers *cfa2040* and *barc182* (position: 164 cM). Ren and co-workers reported a 36.5 ± 6.75 cM recombination distance between *Yr39* and *Yr52* [24]. Another HTAP resistance gene, *Yr59*, was located between *cfa2040* and *barc182* and Zhou and co-workers [25] estimated a recombination fraction of 5.4 ±7.6 cM between *Yr52* and *Yr59*. These workers also placed *YrC591* distal to *Yr59* (12.3 cM) based on common markers. Different genetic maps and recombination values reported by various workers demonstrated that *YrVL2/YrC591* represented a new locus; it was permanently named *Yr67*.

The closely linked markers for *Yr66* and *Yr67* showed more than 85% polymorphism among a set of 89 wheat cultivars. These results indicated the usefulness of *Yr66*- and *Yr67*-linked markers for the marker-assisted selection of these genes in breeding programs following the confirmation of polymorphism between the donor and recurrent parents. These genes have broadened the resistance gene pool. Markers for many seedling stripe rust resistance genes, including *Yr47* [14], *Yr51* [26], *Yr57* [27], *Yr63* [5], *Yr81* [28], and *Yr82* [29], as well as adult plant resistance genes *Yr18* [30], *Yr36* [31], and *Yr46* [32], have been reported. Similarly, markers for several leaf rust and stem rust resistance genes have been reported. The markers identified in this study can be used to develop triple-rust-resistant wheat cultivars through marker-assisted resistance gene pyramiding.

## 5. Conclusions

The genetic analysis of stripe rust resistance in wheat cultivar VL Gehun 892 led to the naming of two new stripe rust resistance loci *Yr66* (chromosome 3D) and *Yr67* (chromosome 7B). Markers closely linked with *Yr66* (*KASP_48179* and *KASP_18087*) and *Yr67* (*KASP_2239*) were developed and validated on a set of 90 Australian wheat cultivars. More than 85% of wheat cultivars amplified alleles alternate to those linked with *Yr66* and *Yr67* when tested with these markers. These results supported the use of *Yr66*- and *Yr67*-linked markers for the marker-assisted pyramiding of these genes with other marker-tagged rust resistance loci in wheat improvement programs to achieve the durable control of rust diseases. The marker-assisted selection of other marker-tagged loci that control economic/quality traits in wheat can allow the delivery of the total genotypic package in future wheat cultivars.

**Author Contributions:** H.B., U.B. and L.K. planned the project. H.B., L.K. and H.M. developed the material. H.B., U.B. and H.M. conducted greenhouse screening. U.B., N.Q. and K.F. performed genotyping. H.B., U.B. and N.Q. drafted the manuscript and all authors read it. H.B. and U.B. provided overall supervision. All authors have read and agreed to the published version of the manuscript.

**Funding:** Funding sources are acknowledged in the acknowledgements section.

**Data Availability Statement:** Data are available from the first and last authors.

**Acknowledgments:** We thank AusAID for the Australian Leadership Awards Fellowships (Grant Number 54870) and the Grains Research and Development Corporation (GRDC) Australia (Grant number 9176057) for financial support. Lakshmi Kant is thankful for the management of ICAR-VPKAS, Almora, Uttarakhand, India, for study leave. We thank G.P. Singh (Director, Indian Council of Agricultural Research (ICAR)-Indian Institute of Wheat and Barley Research, Karnal, India) and Richard Trethowan (Acting Director, Plant Breeding Institute, The University of Sydney) for reviewing the manuscript.

**Conflicts of Interest:** The authors declare no conflict of interest.

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
