# Peer review of "Identification and Characterisation of Stripe Rust Resistance Genes Yr66 and Yr67 in Wheat Cultivar VL Gehun 892"

_agronomy, doi:10.3390/agronomy12020318_

Round 1
Reviewer 1 Report
The manuscript contains the mapping results of genes that determine resistance to yellow rust pathogen P. striiformis f. sp. tritici. The authors performed genetic mapping using SNP markers and developed KASP markers that can be used in breeding to identify these resistance genes.
There are comments to the article:
- The Introduction is scanty and does not give a complete understanding of the research problem. It is necessary to expand Introduction, for example, due to information about genes that determine resistance (number of genes, effectiveness against different pathotypes), methods of gene mapping, KASP markers
- Line 34 AUD127 abbreviation, it is necessary to decipher
- Lines 42-45 provide references on avirulence/virulence to known resistance genes
- Lines 50, 66 – decipher infection types IT) ;CN, ITs;1-C to 2C vs. IT3+ (provide additional information in Material and Methods)
- Lines 55 - susceptible a seedling susceptible (??)
- Lines 54-61– this is not the Introduction, transfer to the Material and Methods
- In the section Greenhouse testing, please, add information about how the type of reaction was determined
- Line 92 – provide information about manufacturers of 90K SNP array and references on this SNP markers
- Line 93 - It is not clear how associations were calculated
- Lines 94-95 – provide information about PCR condition for KASP markers and company where these markers were converted
- Chapter 2. Molecular mapping
Information is needed (as table supplements) about the polymorphism of SNP markers for all chromosomes, how many were used, how many polymorphic, how many showed marker-trait associations (MTAs). - Line 114 – correct the title of table 1
- Lines 120-127 Provide several Figures for KASP markers that reveal polymorphism between parental forms
- Table 2. Give a reference to the physical position of SNP markers
- The authors used SSR markers to map resistance genes. Please provide the results (the number of markers and their polymorphism, length of the amplified fragments), may be as supplemental table
- KASP2239 marker is located at a distance of 3.6 cM from the gene, and KASP37096 locates much closer to the gene (1.2 cM). How do the authors explain the absence of polymorphism (<50%)? Please, provide explanation in Discussion
- citation in the text should be done according to the rules of the Agronomy journal
Author Response
We appreciate positive feedback from the reviewer and thank him or her for helping us to improve this manuscript.
The manuscript contains the mapping results of genes that determine resistance to yellow rust pathogen P. striiformis f. sp. tritici. The authors performed genetic mapping using SNP markers and developed KASP markers that can be used in breeding to identify these resistance genes.
There are comments to the article:
- The Introduction is scanty and does not give a complete understanding of the research problem. It is necessary to expand Introduction, for example, due to information about genes that determine resistance (number of genes, effectiveness against different pathotypes), methods of gene mapping, KASP markers
Author response: Suggestion accepted and text modified
- Line 34 AUD127 abbreviation, it is necessary to decipher
Author response: Suggestion accepted and text modifie
- Lines 42-45 provide references on avirulence/virulence to known resistance genes
Author response: Suggestion accepted and text modified
- Lines 50, 66 – decipher infection types IT) ;CN, ITs;1-C to 2C vs. IT3+ (provide additional information in Material and Methods)
Author response: Suggestions accepted and text modified
- Lines 55 - susceptible a seedling susceptible (??)
Author response: Suggestion accepted and text modified
- Lines 54-61– this is not the Introduction, transfer to the Material and Methods
Author response: Suggestion accepted and text modified
- In the section Greenhouse testing, please, add information about how the type of reaction was determined
Author response: Suggestion accepted and text modified
- Line 92 – provide information about manufacturers of 90K SNP array and references on this SNP markers
Author response: Suggestion accepted and text added
- Line 93 - It is not clear how associations were calculated
Author response: Suggestion accepted and text added
- Lines 94-95 – provide information about PCR condition for KASP markers and company where these markers were converted
Author response: Already available in the reference, let us not keep adding it.
- Chapter 2. Molecular mapping
Information is needed (as table supplements) about the polymorphism of SNP markers for all chromosomes, how many were used, how many polymorphic, how many showed marker-trait associations (MTAs).
Author response: It is clearly stated in text and this manuscript doe not cover QTL mapping; it is about two discrete loci.
- Line 114 – correct the title of table 1
Author response: Corrected
- Lines 120-127 Provide several Figures for KASP markers that reveal polymorphism between parental forms
Author response: I do not believe in re-inventing the wheel and presenting the obvious. These figures are not necessary.
- Table 2. Give a reference to the physical position of SNP markers
Author response: It is already covered in discussion.
- The authors used SSR markers to map resistance genes. Please provide the results (the number of markers and their polymorphism, length of the amplified fragments), may be as supplemental table
Author response: No SSR data included and hence section and the reference deleted.
- KASP2239 marker is located at a distance of 3.6 cM from the gene, and KASP37096 locates much closer to the gene (1.2 cM). How do the authors explain the absence of polymorphism (<50%)? Please, provide explanation in Discussion
Author response: The chromosome 7B of wheat is a mess. You can have 7B, 5BS.7BS, 5BL.7BL, etc. This anomaly is due to these rearrangements. We dealt with a discrete locus and let us not worry about the rest.
- citation in the text should be done according to the rules of the Agronomy journal
Author response: citation corrected.

Reviewer 2 Report
In my considered opinion, this is manuscript presents original research that has generated quality scientific findings of significance to inform breeding works on stripe rust resistance in wheat. The authors have used sound methodology and comprehensively presented their findings.
The quality of English language is quite impressive, simple and easily understood.
Author Response
-
